# Explaining Exploration–Exploitation in Humans

Antonio Candelieri [1,*] , Andrea Ponti [1] and Francesco Archetti [2]

1 Department of Economics, Management and Statistics, University of Milano-Bicocca, 20126 Milan, Italy

2 Department of Computer Science, Systems and Communication, University of Milano-Bicocca, 20126 Milan, Italy

* Correspondence: antonio.candelieri@unimib.it

**Abstract:** Human as well as algorithmic searches are performed to balance exploration and exploitation. The search task in this paper is the global optimization of a 2D multimodal function, unknown to the searcher. Thus, the task presents the following features: (*i*) *uncertainty* (i.e., information about the function can be acquired only through function observations), (*ii*) *sequentiality* (i.e., the choice of the next point to observe depends on the previous ones), and (*iii*) *limited budget* (i.e., a maximum number of sequential choices allowed to the players). The data about human behavior are gathered through a gaming app whose screen represents all the possible locations the player can click on. The associated value of the unknown function is shown to the player. Experimental data are gathered from 39 subjects playing 10 different tasks each. Decisions are analyzed in a Pareto optimality setting— improvement vs. uncertainty. The experimental results show that the most significant deviations from the Pareto rationality are associated with a behavior named "*exasperated exploration*", close to random search. This behavior shows a statistically significant association with stressful situations occurring when, according to their current belief, the human feels there are no chances to improve over the best value observed so far, while the remaining budget is running out. To classify between Pareto and Not-Pareto decisions, an explainable/interpretable Machine Learning model based on Decision Tree learning is developed. The resulting model is used to implement a *synthetic human searcher/optimizer* successively compared against Bayesian Optimization. On half of the test problems, the synthetic human results as more effective and efficient.

**Keywords:** active human learning; exploration–exploitation dilemma; explainable machine learning

## 1. Introduction

### 1.1. Motivation

Many human activities concerning decision-making under uncertainty are often addressed through a *trial-and-error process,* in which every sequential decision is taken to balance between two different goals: *exploitation* and *exploration*. While the former refers to using the *knowledge* collected so far to maximize the *immediate reward,* the latter is related to increasing the knowledge about the target problem. The two goals are usually antagonistic, leading to the so-called *exploration–exploitation dilemma* [1,2]. An exemplary task is the *search process,* which is the specific setting considered in this paper. A group of 39 subjects is asked to play the following game, individually and independently: given 20 trials, the player clicks on a 2D white screen, immediately observes the score associated with the choice and continues with the aim to find the maximum possible score, within the given number of trials. The value of the maximum score is not known. The task is characterized by:

- *Uncertainty*—the only knowledge about scores is that collected after making choices;
- *A sequential nature*—according to the score observed so far, the human player updates their beliefs about the possible value and location of the maximum score and then makes the next decision, to balance between trusting their expectation (i.e., exploitation) and fulfilling their need for uncertainty reduction (i.e., exploration);

- Finally, a *limited budget*—a limited number of sequential trials.

It is easy to understand that this setting is nothing more than the global optimization of a black-box function, that is:

$$x^* = \underset{x \in \Omega \subset \Re^d}{\operatorname{argmax}} f(x) \tag{1}$$

where $f(x)$ is, in our setting, the unknown function whose evaluation provides the score associated with each location of the 2D screen, and $\Omega$ is the so-called *search space*, which consists of all the possible locations the player can click on.

This paper stems from recent results of the analysis of the search strategies performed by humans playing the described game [3–5]. More specifically:

- In [3], it was empirically proven that the highest score found by humans at the end of the task is closer to that identified by Bayesian Optimization (BO) [6–8] than those found by other state-of-the-art global optimization methods. This outcome was obtained on one objective function, $f(x)$, and a group of 60 subjects. Indeed, the BO's underlying rationale is reasonable for a human being: an approximation of $f(x)$ is generated according to the observations collected so far—in BO, it is given by a *probabilistic surrogate model*, usually a Gaussian Process (GP) regression model—then the next location is chosen depending on the prediction provided by this approximation and the associated predictive uncertainty. In BO, these two values are combined by an *acquisition function* (aka *utility function* or *infill criterion*), implementing an exploitation–exploration trade-off mechanism. It is important to remark that, although in [3], there were no significant differences on the highest scores found by humans and BO at the end of the task, differences occur with respect to single sequential decisions.

- In [4], the analysis focused on investigating the possible root causes of the previously mentioned differences. GP modeling and Bayesian learning [9,10] emerged as central paradigms in modeling *human learning*, but fitting a GP requires to choose, a priori, a *kernel* that implies specific (shape) features on the approximation of $f(x)$, along with the associated predictive uncertainty. In [11], it was demonstrated that *"GPs with standard kernels struggle on function extrapolation problems that are trivial for human learners"*, but [4] empirically proved (on 14 subjects and 10 different objective functions) that a suitable *uncertainty quantification* measure can offer a sounder explanation, than GP's kernel, of the differences between humans' and BO's single decisions. Indeed, as reported in [12], different quantifications of the uncertainty are at the core of theories of cognition and emotion and can, therefore, significantly affect the exploitation– exploration trade-off mechanism in humans' search. Moreover, in [4], a Pareto analysis was performed, having as objectives the uncertainty quantification and the possible improvement with respect to the best score observed by the human player so far. Empirical results proved that BO acquisition functions are limited and cannot capture all the choices performed by the humans. Indeed, they can only make Pareto optimal decisions (i.e., BO is Pareto-rational), with all their possible exploitation–exploration trade-offs lying on the Pareto frontier. On the contrary, humans can also make decisions that are far away from the Pareto frontier: this specific behavior has been named *"exasperated exploration"* (just to differ from the exploration provided by the BO's acquisition function). It basically coincides with *pure random search* and occurs with high values of the searcher's stress. Intuitively, the stress of a human player is associated with their perception that, depending on the current beliefs, there is no chance to further improve the best score collected so far. Stress increases with an increasing number of non-improving trials and with a decreasing number of remaining trials, and it has been measured in terms of Average Cumulative Reward (ACR). It is important to remark that cumulating score is not the goal of the task assigned to humans; ACR has only been used as a possible index quantifying the human player's stress.

- In [5], the Pareto compliance of the behavior of humans was represented as a discrete distribution binned in deciles, obtaining histograms. The Wasserstein distance was used to measure the similarity between the behavior of different users. This

distributional analysis—related to decisions by 14 subjects over 10 different objective functions—was conducted at the individual level and an aggregate level, computing barycenters and performing clustering in the Wasserstein space. It is also interesting to remark that while most of the previous works in cognitive sciences addressed the issue of how people assess the information value of possible queries, in [5], the issue of the perception of probabilistic uncertainty itself was instead addressed.

This paper aims to move a step further from [3–5] with the aim to obtain an explainable/interpretable Machine Learning model—i.e., a Decision Tree (DT)—of humans' search strategy. The final goal is to find a sort of *"synthetic human searcher/optimizer"* that could improve over a standard BO algorithm. This goal is not unreasonable: human learners are quite effective at balancing exploration and exploitation, and at updating their generalization model as new information is available, also in unfamiliar contexts. Therefore, the characterization of their behavior is an important opportunity for Machine Learning.

### 1.2. Contributions

The main contributions of this paper are:

- Extending the experimental setting in terms of number of subjects (from fourteen to thirty-nine), with respect to our previous works [4,5], with the aim to obtain more general and robust results.
- Providing an explainable/interpretable Machine Learning model (i.e., a Decision Tree) simulating the trade-off between exploration and exploitation performed by humans, including the exasperated exploration behavior.
- Adopting the explainable/interpretable Machine Learning model to implement a "synthetic human searcher" to compare against standard Bayesian Optimization.

### 1.3. Related Works

In Section 1.1, we briefly introduce the issue of uncertainty quantification in humans and its relationship with learning-and-optimization, and new analytical tools to characterize humans' behavior. Here, we provide a more specific analysis of the prior work and significant recent results.

#### 1.3.1. Cognitive Sciences

An early contribution [13] analyzed how humans manage the trade-off between exploration and exploitation in non-stationary environments. Successively, [1] demonstrated that humans use both *random* and *directed exploration*. Another study [14] showed how directed exploration in humans amounts to adding an *"uncertainty bonus"* to estimated reward values. The same approach was elaborated in [15], which distinguished between *irreducible uncertainty*, related to the reward stochasticity, and *uncertainty*, which can be reduced through information gathering. In the former, the decision strategy is *random search*, while for the latter, it is *directed exploration*, which attaches an uncertainty bonus to each decision value. This distinction mirrors the one in Machine Learning between *aleatoric uncertainty*—due to the stochastic variability due to querying $f(x)$—and *epistemic uncertainty*—due to the lack of knowledge about the actual structure of $f(x)$—which can be reduced by collecting more information. Another study [16] analyzed how entropy and expected utility account, respectively, for exploratory and exploitative behaviors, relating them "*to the discrepancy between observed and expected reward, known as the reward prediction error (RPE), which serves as a learning signal for updating reward expectations. On the other hand, dopamine also appears to participate in various probabilistic computations, including the encoding of uncertainty and the control of uncertainty-guided exploration*" [14]. In addition, [17] analyzed in 1D optimization problems how human active search is explained by BO. Moreover, [18] proposed the use of *function learning* as a mechanism for generalizing prior experiences by approximating a global value function over all the possible options, including those not yet experienced.

### 1.3.2. Economics

The issue of deviations from Pareto optimality has become a central topic in behavioral economics, from the seminal work in [19] to [20], which identified the most common causes for dominance violations, specifically: *framing* (i.e., presentation of a decision problem), *reference points* (i.e., a form of prior expectation), *bounded rationality*, and *emotional responses*. An entirely different approach was suggested in [21], where the concept of ergodicity from statistical mechanics was proposed to model a non-Paretian behavior.

Moreover, according to a famous Schumpeter quotation [22], traditional decision-making under risk *"has a much better claim to being called a logic of choice than a psychology of value"* and, indeed, deviations from Pareto rational behavior have been documented in domains such as economics, business, but also Reinforcement Learning. The analysis of violations of dominance in decision-making has become mainstream economics under the name of *behavioral economics* and *prospect theory* [23]: rather than being labeled "irrational", the non-Pareto-compliant behavior is just not well described by the rational-agent model. Finally, the analysis of the strategies implemented by humans in dealing with uncertainty has been an active research topic [12,24,25]: a key motivation of this line of research is the awareness that *human learners* are amazingly fast and effective at adapting to unfamiliar environments and incorporating upcoming knowledge: this is an intriguing behavior for cognitive sciences as well as an important challenge for Machine Learning.

### 1.3.3. Bayesian Optimization

In the BO research community, recent papers proposed the analysis of the exploration–exploitation dilemma as a bi-objective optimization problem: minimizing the predictive mean (associated with exploitation) while maximizing uncertainty, typically the predictive standard deviation (associated with exploration) [26]. This multi-objective approach is suitable also for the *Upper Confidence Bound* (UCB) acquisition function in Multi-Armed Bandits [27] and BO [28]. This mean-variance framework was also considered in [29], for multi-task, multi-objective, and constrained optimization scenarios. In addition, [30,31] showed that taking a decision by randomly sampling from the Pareto frontier can outperform other acquisition functions. The main motivation is that the Pareto frontier offers a set of Pareto-efficient decisions wider than that allowed by "traditional" acquisition functions. A recent important contribution is [32], which tackled the problem to infer, given the observed search path generated by a human subject in the execution of a black box optimization task, the unknown acquisition function underlying the sequence. For the solution of this problem, also known as Inverse Bayesian Optimization (IBO), a probabilistic framework for the non-parametric Bayesian inference of the acquisition function is proposed, given a set of possible acquisition functions.

### 1.3.4. Synthetic Human Searcher/Optimizer

*Artificial human optimization* has already received some attention and a new human-driven optimizer has been proposed by modifying PSO (Particle Swarm Optimization) with concepts based on human traits such as kindness, sickness, and relaxation [33,34]. Modulating these traits brings about different specializations of PSO and can be seen as a relatively unprincipled way to manage exploration vs. exploitation. On the contrary, BO offers a unified theory based on the general model of Bayesian learning, which has been shown to be closer to the human behavior.

Another line of research connected to the synthetic human optimizer is related to "Interactive BO" [35,36]: in this case, the objective is the global optimization of experiments with multiple attributes, aimed at accommodating partial preference information. An algorithm interacts sequentially with a human decision-maker while optimizing a time-consuming-to-evaluate objective function. By leveraging preference information, the approach is more effective than multi-objective optimization, which typically does not use interaction with the human to focus optimization on those parts of the Pareto frontier most likely to contain the human-preferred information.

### 1.4. Outline of the Paper

The rest of the paper is organized as follows: Section 2 summarizes the details about data collection, subjects involved in the study, and methods adopted in the analysis. Section 3 reports the experimental results and the most relevant considerations. Section 4 focuses on the comparison between the synthetic human and traditional Bayesian Optimization. Finally, Section 5 summarizes the most relevant conclusions, the limitations, and the research perspectives on the topic.

## 2. Materials and Methods

### 2.1. The Gaming App

A web-based application was developed to implement the search game and to collect human players' decisions. The latest version of the gaming app was presented in [5] and is reported here in Figure 1. Every human player is asked to search for the highest score over a 2D panel (i.e., the game field), given 20 sequential trials. After each click, the corresponding score is shown to the player so that they can update their own belief about the underlying scoring mechanism (i.e., $f(x)$). Color and size of each selected point are correlated to the associated score, enabling an easier understanding. Some relevant information about the gameplay is also provided, specifically, the highest score observed so far and the remaining trials (aka "remaining shots").

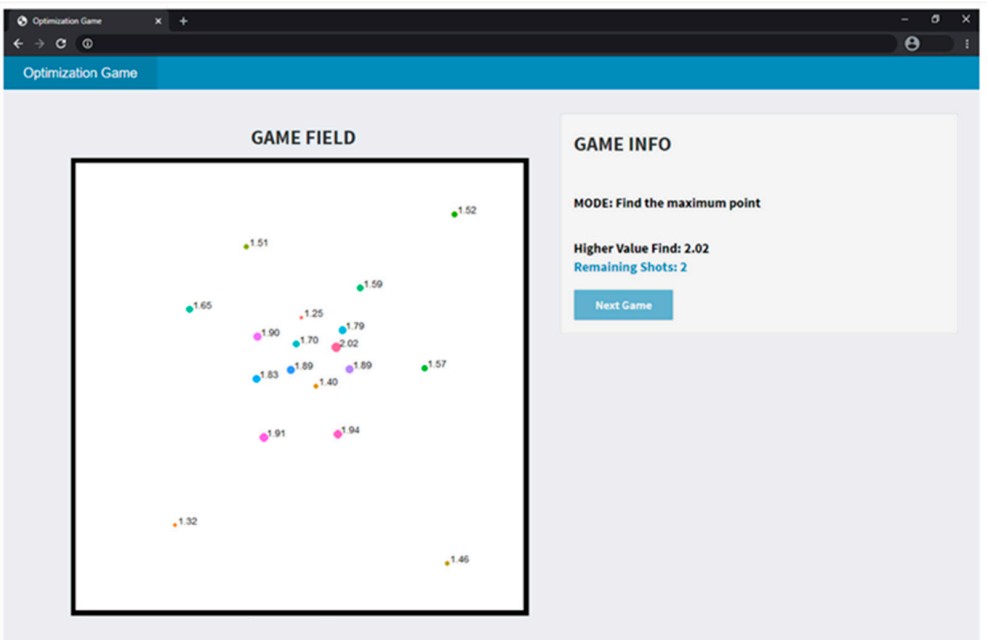

**Figure 1.** Web gaming app: an example of gameplay (source [5]).

### 2.2. Involved Subjects

Thirty-nine (39) subjects, 14 female (25 male), voluntarily and independently decided to sign-in and play our game, which was made available on Amazon Mechanical Turk. Among them, the level of education was: 26 bachelor's degree, 9 master's degree, 3 diploma, and 1 higher. As far as their occupation is concerned, 34 were employees, 2 were students, 1 was unemployed, and 2 indicated "other".

Each subject played 10 different test functions (reported in Appendix A), properly chosen to have a sufficiently large set of differently shaped functions. It is easy to notice that some of them were more difficult globally optimize than others.

### 2.3. Modeling: Gaussian Process Regression

A Gaussian Process (GP) is a random distribution over functions and is denoted with $f(x) \sim GP(\mu(x), k(x, x'))$, where $\mu(x) = \mathbb{E}(f(x)) : \Omega \to \Re$ is the mean function of the GP and $k(x, x\prime) : \Omega \times \Omega \to \Re$ is the kernel (aka covariance) function. In Machine Learning, GP modeling is largely used for both classification and regression tasks [37,38], providing probabilistic predictions by conditioning $\mu(x)$ and $\sigma^2(x)$ on a set of available data/observations. Let us denote with $X_{1:n} = \left\{ x^{(i)} \right\}_{i=1,\dots,n}$ a set of $n$ data points in $\Omega \subset \Re^d$, and with $y_{1:n} = \left\{ f\left(x^{(i)}\right) + \varepsilon \right\}_{i=1,\dots,n}$ the associated function values, possibly noisy with $\varepsilon$ being zero-mean Gaussian noise: $\varepsilon \sim \mathcal{N}(0, \lambda^2)$. Then, the GP's posterior predictive mean and standard deviation, conditioned on $X_{1:n}$ and $y_{1:n}$, are, respectively, given by:

$$\mu(x) = \mathrm{k}(x, X_{1:n}) \left[ \mathrm{K} + \lambda^2 I \right]^{-1} y_{1:n} \tag{2}$$

$$\sigma^2(x) = k(x, x) - \mathrm{k}(x, X_{1:n}) \left[ \mathrm{K} + \lambda^2 I \right]^{-1} \mathrm{k}(X_{1:n}, x) \tag{3}$$

where $\mathrm{k}(x, X_{1:n}) = \left\{ k\left(x, x^{(i)}\right) \right\}_{i=1,\dots,n}$ and $\mathrm{K} \in \Re^{n \times n}$ is the kernel matrix, with entries $\mathrm{K}_{ij} = k\left(x^{(i)}, x^{(j)}\right)$.

The choice of the kernel establishes prior assumptions over the structural properties (i.e., smoothness) of the underlying $f(x)$. Almost every kernel has its own hyperparameters to tune—usually via Maximum Log-likelihood Estimation (MLE) or Maximum A-Posteriori (MAP)—for reducing the mismatches between the assumed smoothness and the data. Common kernels for GP regression—considered in this paper—are:

- Squared Exponential: $k_{SE}(x, x') = e^{-\frac{||x-x'||^2}{2\ell^2}}$;
- Exponential: $k_{EXP}(x, x') = e^{-\frac{||x-x'||}{\ell}}$;
- Power-exponential: $k_{PE}(x, x') = e^{-\frac{||x-x'||^p}{\ell^p}}$;
- Matérn3/2: $k_{M3/2}(x, x') = \left(1 + \frac{\sqrt{3}\,||x-x'||}{\ell}\right) e^{-\frac{\sqrt{3}\,||x-x'||}{\ell}}$;
- Matérn5/2: $k_{M5/2}(x, x') = \left[1 + \frac{\sqrt{5}\,||x-x'||}{\ell} + \frac{5}{3}\left(\frac{||x-x'||}{\ell}\right)^2\right] e^{-\frac{\sqrt{5}\,||x-x'||}{\ell}}$.

With $\ell$ as the so-called *length-scale*, it regulates the non-linear relation between the $y$ values associated with two data points, $x$ and $x'$, depending on their distance, $||x - x'||$. When $\ell$ is a scalar, the kernel is said to be *isotropic*; otherwise, $\ell \in \Re_+^d$, modeling a different relation along each direction. Another typical kernel's hyperparameter is a multiplier— usually denoted with $\sigma_f^2$, regulating the variation in amplitude.

### 2.4. Pareto Analysis: Expectation vs. Uncertainty

In our analysis, after each human choice, a GP is trained on the locations and scores collected so far. Then, the next decision performed by the human, namely $x^{(n+1)} \in \Omega$, is mapped into the 2-dimensional space spanned by the following two objectives, $\zeta(x)$ and $z(x)$, both to be maximized:

$$\zeta(x) = \mu(x) - y^+ \tag{4}$$

$$z(x) = \begin{cases} 0 & \text{if } \exists\, x^{(i)} \in X_{1:n} : \left|\left| x - x^{(i)} \right|\right|^2 = 0 \\ \frac{2}{\pi} \tan^{-1}\left( \frac{1}{\sum_{j=1}^n w_j(x)} \right) & \text{otherwise} \end{cases} \tag{5}$$

with $y^+$ being the highest score observed so far, and $w_j(x) = \frac{e^{-||x-x^{(j)}||^2}}{||x-x^{(j)}||^2}$.

The first objective, $\zeta(x)$, represents the expectation that the human player has in improving the current best score, $y^+$, while the second objective, $z(x)$, is an uncertainty quantification measure [39]. Pareto rationality was the theoretical framework used to analyze humans' choices in this bi-objective setting. In multi-objective optimization problems, $q$ objective functions $\gamma_1(x), \ldots, \gamma_q(x)$, with $\gamma_i(x) :\to \Re$, are to be simultaneously optimized in $\Omega \subseteq \Re^d$. We use the notation $\boldsymbol{\gamma}(x) = \left(\gamma_1(x), \ldots, \gamma_q(x)\right)$ to refer to the vector of all objectives evaluated at a location $x$. The goal is to identify the Pareto frontier of $\boldsymbol{\gamma}(x)$. To do this, we need an ordering relation in $\Re^q : \boldsymbol{\gamma} = (y_1, \ldots, y_q) \preccurlyeq \boldsymbol{\gamma}' = \left(y'_1, \ldots, y'_q\right)$ if and only if $\gamma_i \leq \gamma'_i$ for $i = 1, \ldots, q$. This ordering relation induces an order in $\Omega : x \preccurlyeq x'$ if and only if $\boldsymbol{\gamma}(x) \preccurlyeq \boldsymbol{\gamma}(x')$. We also say that $\gamma'$ dominates $\gamma$ (strongly if $\exists\, i = 1, \ldots, q$ for which $\gamma_i < \gamma'_i$). The optimal non-dominated solutions lie on the so-called Pareto frontier. The interest in finding locations $x$ having the associated $\boldsymbol{\gamma}(x)$ on the Pareto frontier is clear: they represent the trade-off between conflicting objectives and are the only ones, according to the Pareto rationality, to be considered.

In our specific case, $q = 2$, with $\gamma_1(x) = \zeta(x)$ and $\gamma_2(x) = z(x)$. Both objectives are not expensive to evaluate; therefore, the Pareto frontier can be easily approximated by considering a fine grid of locations in $\Omega$ without the need to resort to methods approximating expensive Pareto frontiers within a limited number of evaluations.

More precisely, we approximate our Pareto frontier by sampling a grid of $m$ points in $\Omega$, denoted by $\hat{\mathbf{X}}_{1:m} = \left\{ x^{(j)} \right\}_{j=1,\ldots,m}$, and then computing the associated objectives pairs $\Psi_{1:m} = \left\{ \left( \zeta\left(x^{(j)}\right), z\left(x^{(j)}\right) \right) \right\}_{j=1,\ldots,m}$. Therefore, the Pareto frontier can be approximated as:

$$\mathcal{P}(\Psi_{1:m}) = \{\psi \in \Psi_{1:m} : \forall\, \psi' \in \Psi_{1:m}\ \psi \succ \psi'\} \tag{6}$$

where $\psi = (\zeta(x), z(x))$ and $\psi' = (\zeta(x'), z(x'))$, and $\psi \succ \psi' \Leftrightarrow \zeta(x) > \zeta(x') \wedge z(x) > z(x')$.

Every next decision, $x^{(n+1)}$, can be analyzed according to the distance of its "image" $\left( \zeta\left(x^{(n+1)}\right), z\left(x^{(n+1)}\right) \right)$ from the Pareto frontier, computed as follows:

$$d\left(\overline{\psi}, \overline{\mathcal{P}}\right) = \min_{\psi \in \overline{\mathcal{P}}} \left\{ \left\| \overline{\psi} - \psi \right\|_2^2 \right\} \tag{7}$$

where $\overline{\psi} = \left( \zeta\left(x^{(n+1)}\right), z\left(x^{(n+1)}\right) \right)$ and $\overline{\mathcal{P}} = \mathcal{P}(\Psi_{1:m}) \cup \{\overline{\psi}\}$.

Characterizing every decision as Pareto or not-Pareto requires the definition of a suitable threshold on the distance between the decision and the current Pareto front. Indeed, from a theoretical point of view, a decision is Pareto-optimal if it lies exactly on the Pareto front; however, numerical approximation must be considered in practical experiments. From empirical evaluations, we decided to set this threshold to $10^{-4}$, which proved to be suitable with respect to both computational approximation and invariance to all the functions' codomains. In fact, according to the chosen threshold, the number of decisions detected as Pareto did not increase from original distance values and [0, 1]-rescaled values, as summarized in Table 1. For completeness, we clarify that the same invariance effect was also observed for a threshold $10^{-2}$, but we preferred to use $10^{-4}$ to gain a better numerical approximation in computations.

**Table 1.** Percentage of decisions labeled as Pareto (depending on a threshold $10^{-4}$), separately for original and [0, 1]-rescaled distance from Pareto front.

| Test Function | Original Distance from Pareto Front | [0, 1]-Rescaled Distance from Pareto Front |
|---|---|---|
| ackley | 57.29% | 58.46% (+1.17%) |
| beale | 10.15% | 31.06% (+20.91%) |
| branin | 8.96% | 10.59% (+1.63%) |
| bukin6 | 11.79% | 14.39% (+2.6%) |
| goldprsc | 13.51% | 14.09% (+0.58%) |
| griewank | 25.97% | 27.51% (+1.54%) |
| levy | 18.58% | 22.11% (+3.53%) |
| rastr | 23.00% | 25.11% (+2.11%) |
| schwef | 95.26% | 95.26% (+0.00%) |
| stybtang | 14.95% | 17.79% (+2.84%) |

*2.5. Decision Tree Analysis*

A Decision Tree (DT) is a hierarchical, tree-like structure that considers all the available observations (aka training set) at the root node of the tree. Learning is based on a *divide-et-impera* approach: initially, the observations belonging to the root node are split into (usually two) branches, depending on a suitable value of one of the variables (aka feature) representing the observations. The choice of the most suitable variable and value is based on a measure of "purity" of the "children nodes" resulting from splitting, where "purity" intuitively means that the considered node has to contain as many observations belonging to only one class (aka group) as possible. The splitting process is then iterated over (leave) nodes having a small purity, with the aim to generate, hierarchically, new, more pure nodes. A node containing only observations belonging to a class is pure and does not need to be further split. A common choice for the purity measure is the Information Gain, that is the difference between entropy in the parent and in the children nodes. One of the main advantages of a DT is that the resulting model is human-understandable, a crucial property for the aim of this paper.

**3. Experimental Results**

In this section, we present the most relevant results from this study. First, we perform a Pareto-analysis of the human players' strategies, by comparing the results with those of our previous studies [4,5], due to the larger size of the subjects group considered in this paper. Then, we focus on the outcomes resulting from the application of DT learning to obtain an explainable/interpretable model summarizing a general optimal-search strategy that could be used to implement a *synthetic human searcher/optimizer*, basically an algorithm that could perform global optimization mimicking the strategies adopted by a human player.

*3.1. Pareto Analysis and Comparison with Previous Studies*

According to the threshold defined in Section 2.4, every decision performed by every human player is labeled as "Pareto" (i.e., the decision is closer than $10^{-4}$ from the approximated Pareto front) or "Not-Pareto" (i.e., the decision is farther than $10^{-4}$ from the approximated Pareto front).

Figure 2 shows, for every test problem, the number of human players (on the *y*-axis) with respect to the percentage of Pareto decisions over the 20 available clicks. It is evident that some specific functions induce Pareto decisions (i.e., *ackley, griewank, schwef,* and partially *levy*). This basically confirms our previous results on a smaller group of players.

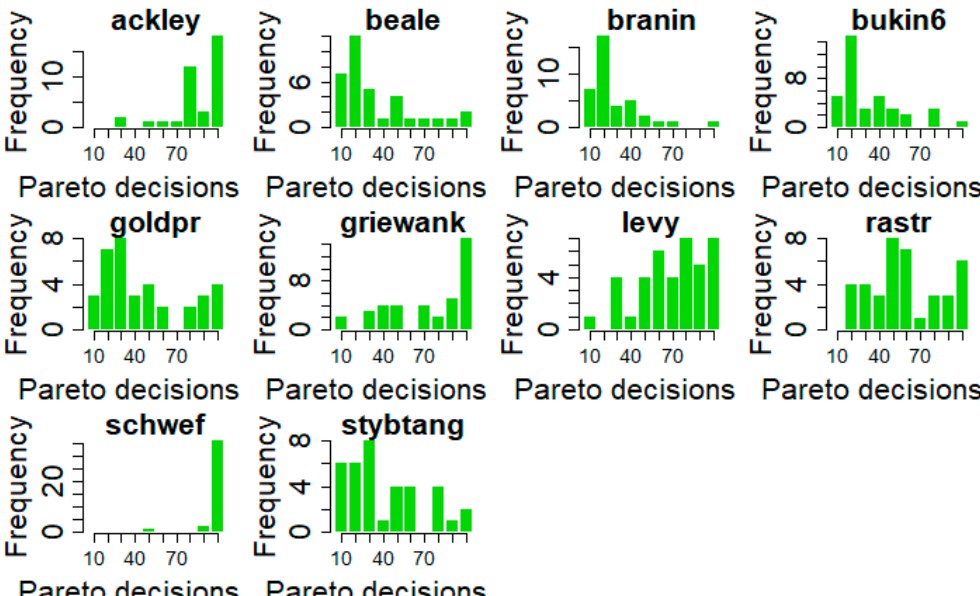

**Figure 2.** Number of human players (*y*-axis) with respect to the percentage of Pareto decisions.

When these histograms are analyzed through an appropriate distance, specifically the Wasserstein distance—as previously performed in [5]—it clearly comes out that the test problems addressed by the human players can be divided into the following two clusters:

- Cluster_1 = (*ackley, griewank, levy, rastr, schwef*);
- Cluster_2 = (*beale, branin, bukin6, goldprsc, stybtang*).

where, as mentioned, Cluster_1 contains the search tasks inducing a higher number of Pareto decisions, while Cluster_2 contains search tasks more associated with the *exasperated exploration* behavior.

Another investigation is related to the metric proposed to quantify the level of stress, possibly leading to the exasperated exploration. The metric is the Average Cumulative Reward (ACR) [4]:

$$ACR^{(t+1)} = \frac{1}{t} \sum_{i=1}^{t} y^{(i)} \tag{8}$$

with $y^{(i)}$ being the score collected according to the *i*-th decision.

From Table 2, it is evident that, on three test problems (i.e., *ackley, goldprsc,* and *schwef*), the ACR cannot be used to differentiate between Pareto and Not-Pareto decisions. However, for the other cases, a statistically significant difference exists. More precisely:

- On 4 cases (i.e., *branin, bukin6, levy,* and *rastr*), Pareto decisions seem to be induced by a higher value of *ACR*. It is important to remark that these 4 cases belong to both the clusters of functions, meaning that a switch between the two types of decisions (Pareto/Not-Pareto) is always possible, at each time-step, depending on the *ACR*.
- On 3 cases (i.e., *beale, griewank,* and *stybtang*), Pareto decisions seem to be induced by a lower value of *ACR*. In addition, in these cases, the test functions belong to both clusters of functions, as remarked in the previous point.

**Table 2.** ACR with respect to Pareto and Not-Pareto decisions, separately for each test function.

| Test Function | ACR Pareto Mean (SD) | ACR Not-Pareto Mean (SD) | U Mann-Whitney Test *p*-Value |
|---|---|---|---|
| ackley | −182.6841 (94.268) | −183.1297 (90.2213) | 0.7447 |
| beale | −98,706.7900 (152,129.9000) | −45,552.8900 (84,082.4400) | **0.0318** |
| branin | −316.1284 (235.0969) | −460.6150 (296.6918) | **<0.001** |
| bukin6 | −694.5507 (410.6313) | −1045.3670 (466.4098) | **<0.001** |
| goldprscsc | −27.2690 (19.5577) | −26.6695 (16.9001) | 0.9825 |
| griewank | −9.8474 (5.2752) | −7.68593 (5.25079) | **<0.001** |
| levy | −92.9790 (70.8990) | −135.2082 (88.2782) | **<0.001** |
| rastr | −298.4032 (166.3426) | −387.9695 (170.5657) | **<0.001** |
| schwef | −8823.6370 (4015.2560) | −9797.6580 (3172.1390) | 0.1772 |
| stybtang | 135.8438 (155.6749) | 244.7435 (244.6132) | **<0.001** |

The results obtained here (Figure 3), on a larger group of subjects, are partially in contrast with previous outcomes [4,5], motivating a deeper analysis and, specifically, the adoption of a multi-variate and human-understandable data analysis approach (i.e., Decision Tree learning).

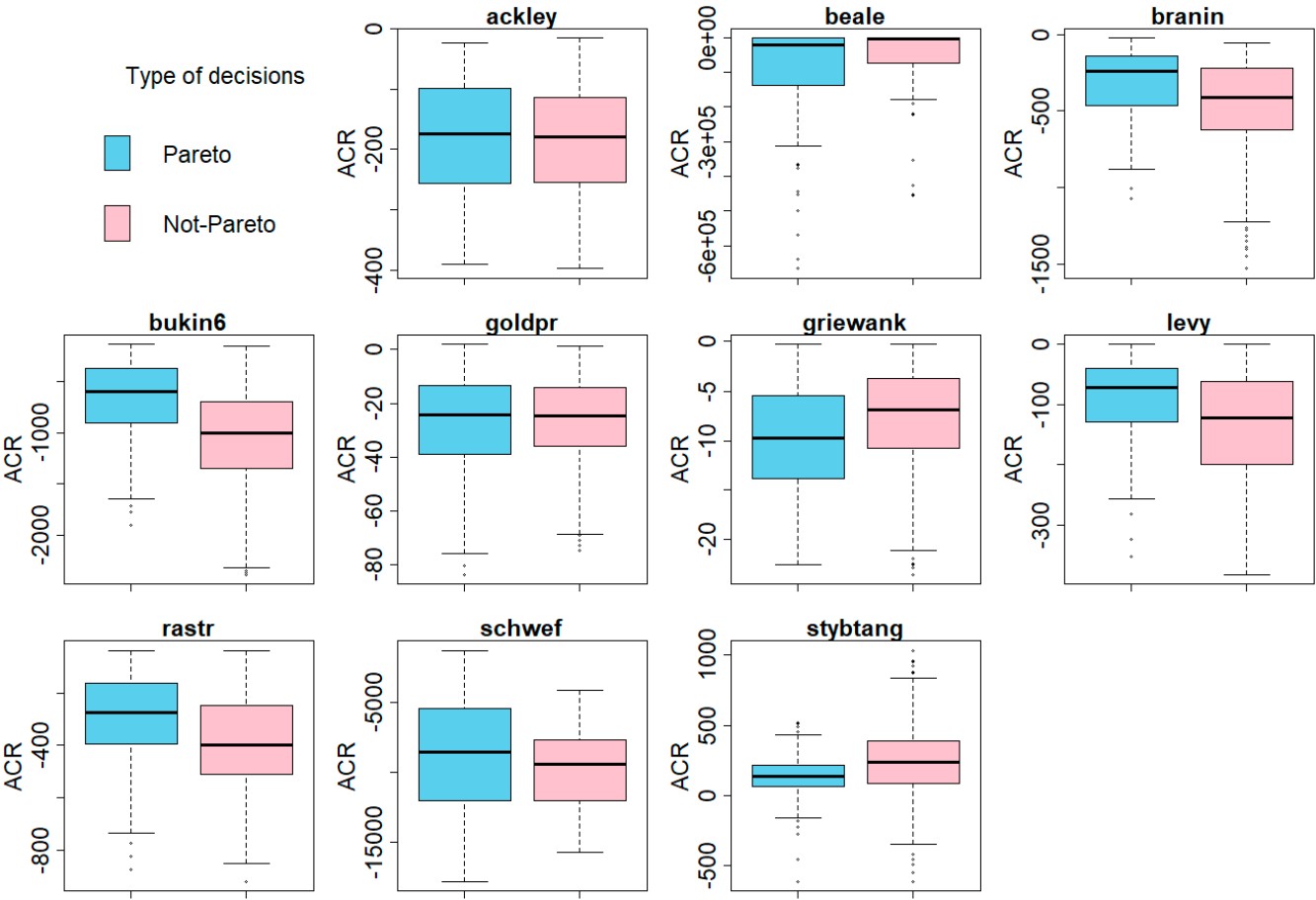

**Figure 3.** ACR with respect to Pareto and Not-Pareto decisions, separately for each test function.

*3.2. Making Humans' Strategies Explainable through Decision Tree Learning*

In this section, we investigate, more deeply, the possible reasons underlying the "exasperated exploration" behavior—which can be basically assimilated to Pure Random Search. As we desire that these reasons could be interpretable, we decide to use a Decision Tree classifier (i.e., J48 from the suite Weka) [40]. The dataset consists of all the choices sequentially performed by the players over their games (excluding the first three "shots" for each player and for each game, because they are obviously explorative). Thus, each row (aka instance) of the dataset is a decision. The columns (aka features) of the dataset are:

1. The black-box function underlying the game—*tf*;
2. The user identifier—*user*;
3. The iteration at which the decision has been taken—*iter*;
4. The ACR—*ACR*;
5. The type of decision, Pareto/not-Pareto, that is the "class label" to be predicted by the Decision Tree.

When all the features are used, the resulting DT provides a very high Accuracy (i.e., percentage of correctly classified decisions) on the entire dataset, that is **training-Accuracy** = 93.07%, but a lower Accuracy on 10-fold-cross validation (i.e., a validation procedure to estimate the accuracy in classifying decisions not into the dataset), that is **10FCV-Accuracy** = 85.45%. The difference between the two accuracy values suggests a slight *overfitting*, meaning that the resulting DT can classify very well the decisions into the dataset, but it is not able to *generalize* on new, previously unseen, decisions. The important outcome is that the most important features to discriminate between Pareto and Not-Pareto decisions are the *test function* (i.e., root node of the DT), and then the *human player*. This remarks that every decision is strongly influenced by the difficulty of the addressed problem and by personal attitude in balancing between exploration and exploration. The DT is too wide to be depicted with a sufficient quality.

As our goal is to search for a DT that can model an as-general-as-possible search strategy, we first manually remove the feature associated with the human player, with the aim to try to first generalize over the different attitudes. The resulting DT shows similar **training-Accuracy** and **10FCV-Accuracy**, 78.61% and 75.86%, respectively. Although overfitting is reduced, the accuracy is too. Moreover, this DT has a higher probability to misclassify a Pareto decision than the previous one. Specifically, the percentage of Pareto decisions correctly classified (i.e., Recall for Pareto class) is 60% (on 10-fold-cross validation), against 85.8% for the Not-Pareto.

Finally, we also remove the feature associated with the test function, with the aim to generalize on both payers and test functions. Thus, the final DT works on two features only: the ACR and the iteration.

The resulting **training-Accuracy** and **10FCV-Accuracy** are, respectively, 74.70% and 73.07%, quite in line with those of the previous DT. Thus, removing the information related to the test function does not imply a drastic reduction in terms of accuracy. However, the probability to misclassify a Pareto decision further increases (i.e., Recall for Pareto class = 52.1%, Recall for Not-Pareto = 86.2%). This is not a critical issue; we just have to be aware that the synthetic human implementing this DT will be even more prone to exploration than a human player (i.e., instead of predicting/suggesting a Pareto decision, it will predict/suggest a Not-Pareto one, in half of the cases). The resulting DT model is quite compact, easy to understand, and is reported in Figure 4.

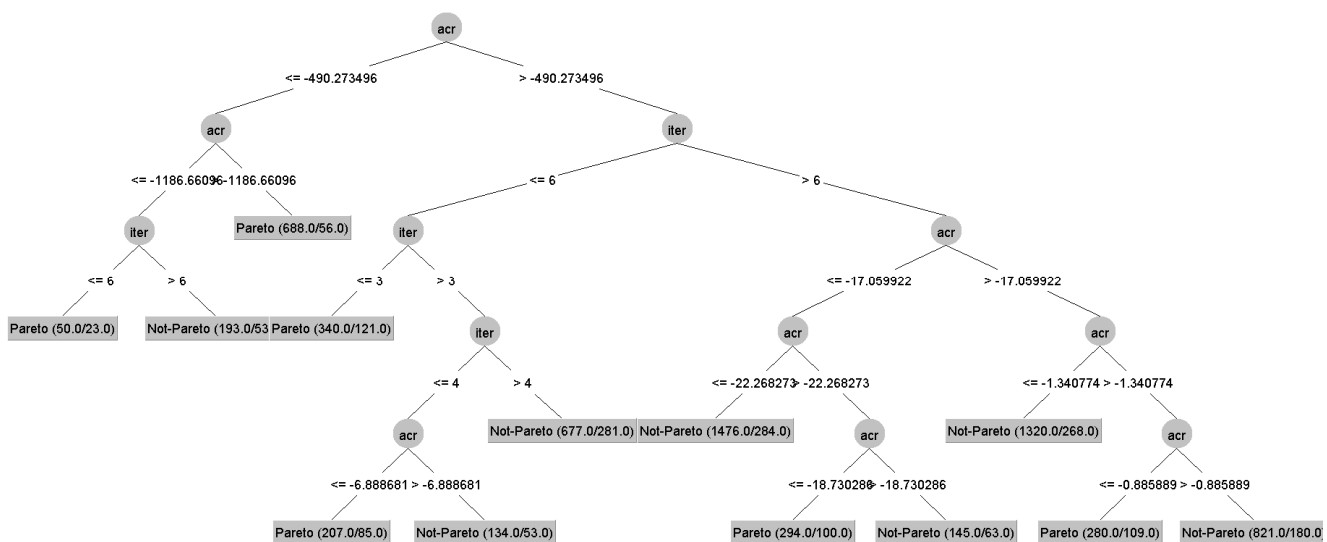

**Figure 4.** DT modeling of the choice between a Pareto and Not-Pareto decision depending on ACR and iteration.

## 4. Testing the Synthetic Human Searcher/Optimizer

As a final experiment, we use the DT reported in the previous section to implement our synthetic human searcher/optimizer and compare it against BO. For each test function, three decisions are initially chosen randomly, then the DT sequentially suggests if the next decision must be Pareto or Not-Pareto. In the first case, just one BO iteration is performed to obtain the next location to query; otherwise, Pure Random Search is adopted, by uniformly sampling the next location to query within the search space.

To mitigate the effect of randomness in the initialization, 30 different independent initializations are considered for each test function. For every run, BO and the synthetic human (SH) share the same set of three initial random queries. Finally, BO uses GP regression with the squared exponential kernel and UCB as the acquisition function. The BO step within the SH uses the same setting.

As relevant results, we observe that BO and SH show significantly similar performances on two test functions, namely "ackley" and "bukin". Figure 5 shows the best score collected up to a specific number of sequential decisions, separately for BO and our SH: solid lines represent the average over the 30 independent runs, while the shaded areas are the standard deviations.

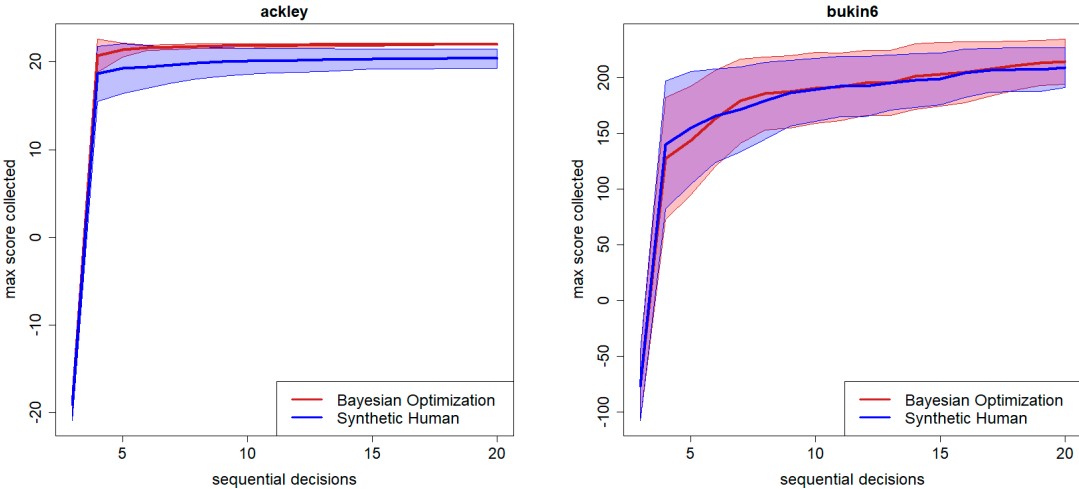

**Figure 5.** Max score observed over sequential decisions: cases with BO and SH offering similar performances.

Then, Figure 6 shows three test functions—namely, "beale", "griewank", and "schwef"—on which BO significantly outperforms our SH. Again, solid lines represent the best score (averaged over 30 independent runs) collected up to a specific number of sequential decisions. Shaded areas are standard deviations. On these three test functions, BO is able to collect a significantly higher best score than our SH, and within a lower number of sequential decisions.

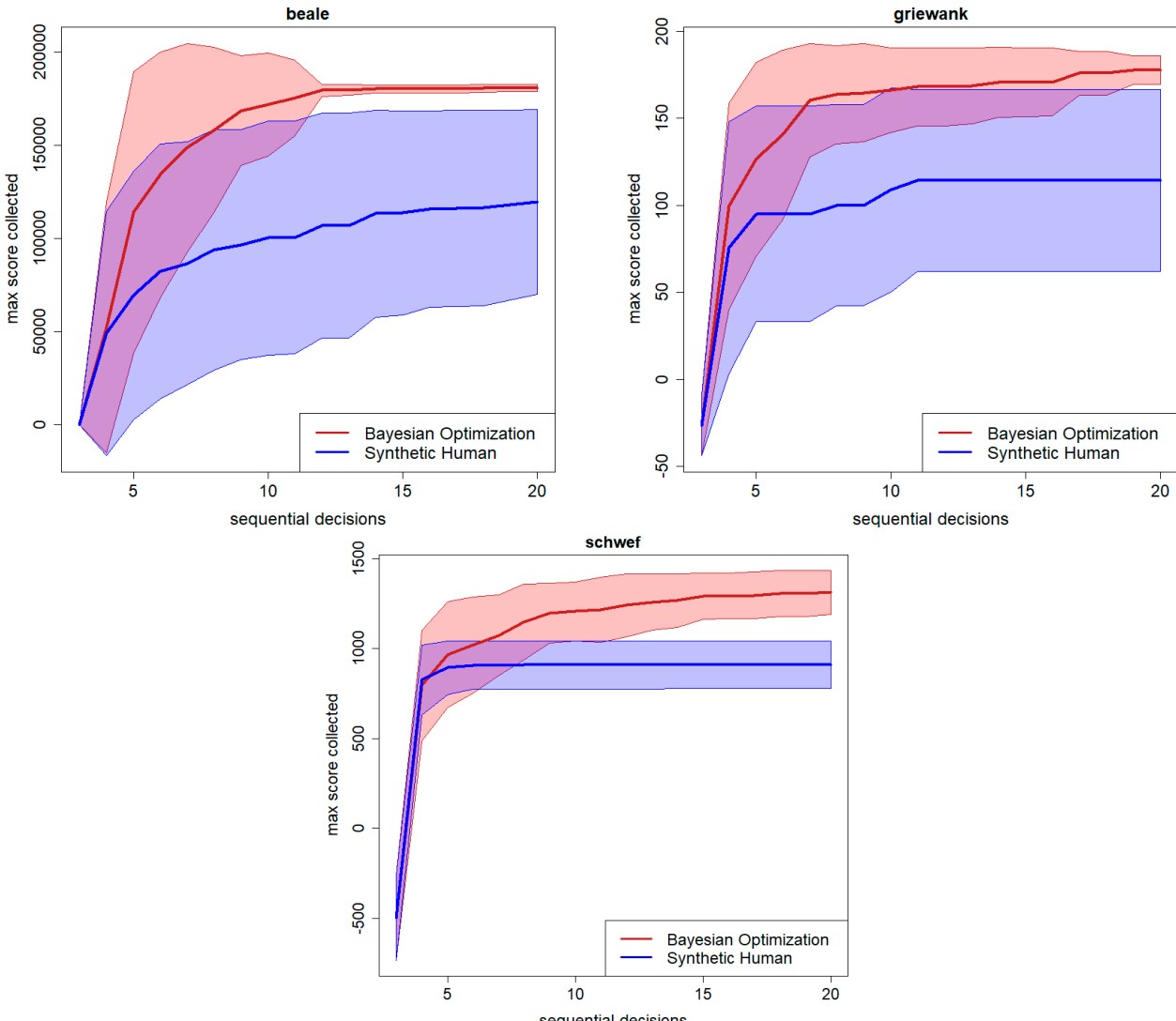

**Figure 6.** Max score observed over sequential decisions: cases with BO outperforming SH.

Finally, Figure 7 shows the remaining five test functions on which our SH significantly outperforms BO. As in the previous charts, solid lines represent the best score (averaged over 30 independent runs) collected up to a specific number of sequential decisions. Shaded areas are standard deviations. Thus, on five out of ten test functions (50%), our SH is able to collect a significantly higher best score than BO, and within a lower number of sequential decisions. The two methods result as significantly similar on two out ten test functions (20%) and BO is significantly better than our SH on three out of ten test functions (30%).

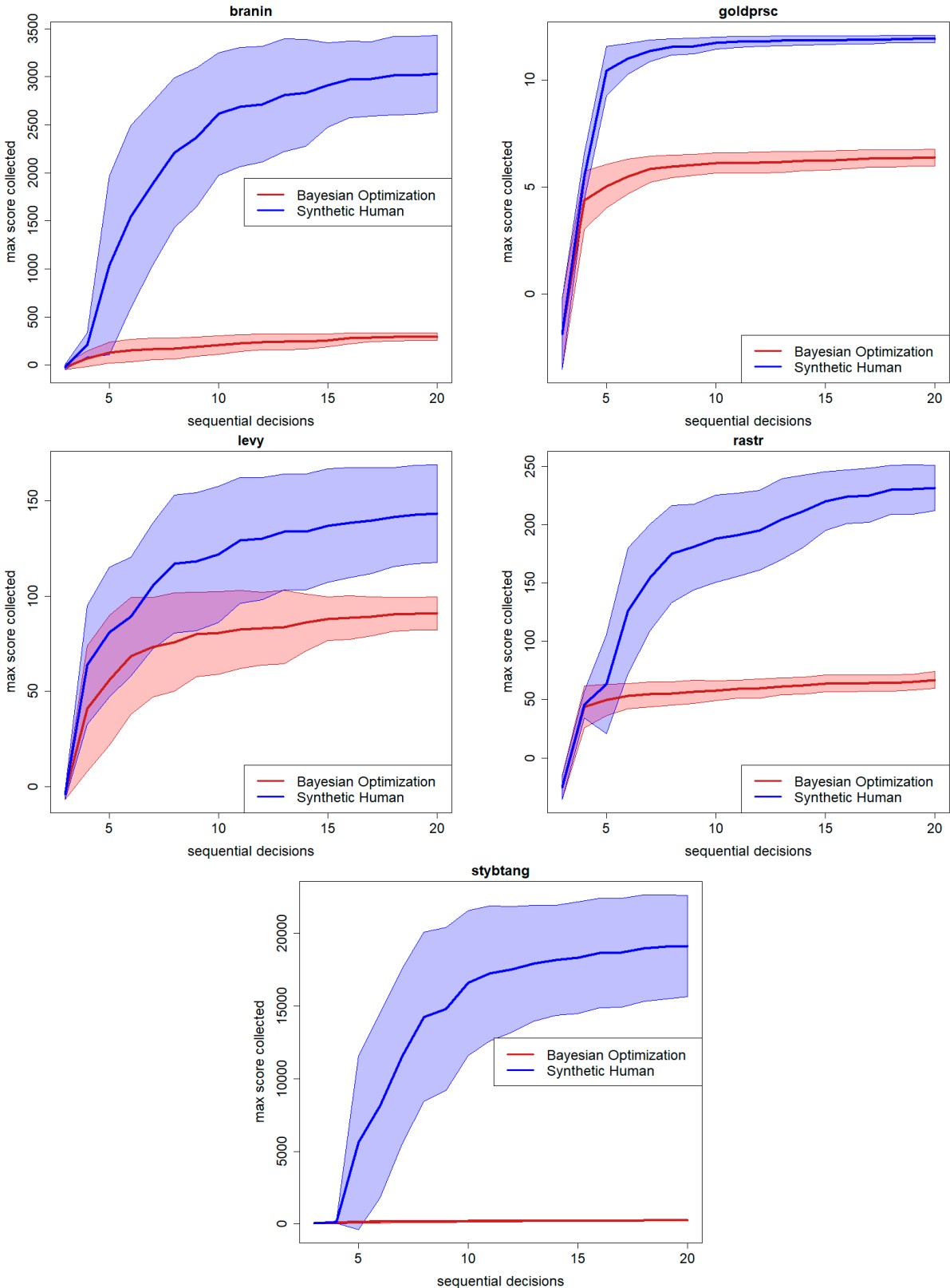

**Figure 7.** Max score observed over sequential decisions: cases with SH outperforming BO.

Thus, the results empirically show that humans are amazingly fast and effective—even more than BO—at adapting to unfamiliar environments and incorporating upcoming knowledge.

## 5. Conclusions

The results of the paper empirically confirm the idea that humans can perform remarkably better than Bayesian Optimization and Machine Learning algorithms at processing new data and experiences to generating new cognitive models and decision-making strategies. This is also remarked by cognitive sciences studies, proving that human decision makers show an effective mechanism for balancing between exploration and exploitation.

More important, this paper also proposes an interpretable and explainable model of the humans' decision process, with respect to the optimization of the specific test functions considered.

A limitation of the study is that the synthetic human searcher/optimizer is based on ACR values related to the set of test functions adopted and, therefore, cannot be considered "general". Indeed, the attitude of the player (risk-averse or risk-seeker) would be affected by their age, education, and gaming experience. However, we cannot include these predictors given the relatively small number of involved subjects. We wish to overcome this limitation by collecting more players' data in the future.

On the other hand, the results show that, given a new problem, few human decisions are sufficient for obtaining a sufficiently effective and efficient synthetic human searcher/optimizer for that specific problem. This consideration is important with respect to the possible application of this approach in business applications, for instance, neuro-marketing. Indeed, a synthetic human can generalize over customer data, available from surveys and online channels, to select the most suitable marketing stimuli to optimize model consumers' cognitive and emotional responses.

**Author Contributions:** Conceptualization, all authors; methodology, A.C. and F.A.; software, A.P.; validation, A.P. and A.C.; data curation, A.P.; writing—original draft preparation, all authors; writing—review and editing, all authors. All authors have read and agreed to the published version of the manuscript.

**Funding:** This research received no external funding.

**Data Availability Statement:** Both data and code for reproducing the analysis and results of this paper are available at the following link: https://github.com/acandelieri/humans_strategies_analysis (accessed on 28 October 2022).

**Acknowledgments:** We greatly acknowledge the DEMS Data Science Lab, Department of Economics Management and Statistics (DEMS), for supporting this work by providing computational resources.

**Conflicts of Interest:** The authors declare no conflict of interest.

## Appendix A

The ten global optimization test functions used in this study, including their analytical formulations, search spaces, and information about optimums and optimizers, can be found at the following link: https://www.sfu.ca/~ssurjano/optimization.html (accessed on 28 October 2022).

Their original formulations refer to minimization problems; to translate them into a maximization problem, the web-based gaming app returns $-f(x)$ as the score (leading to the functions depicted in Figure A1).

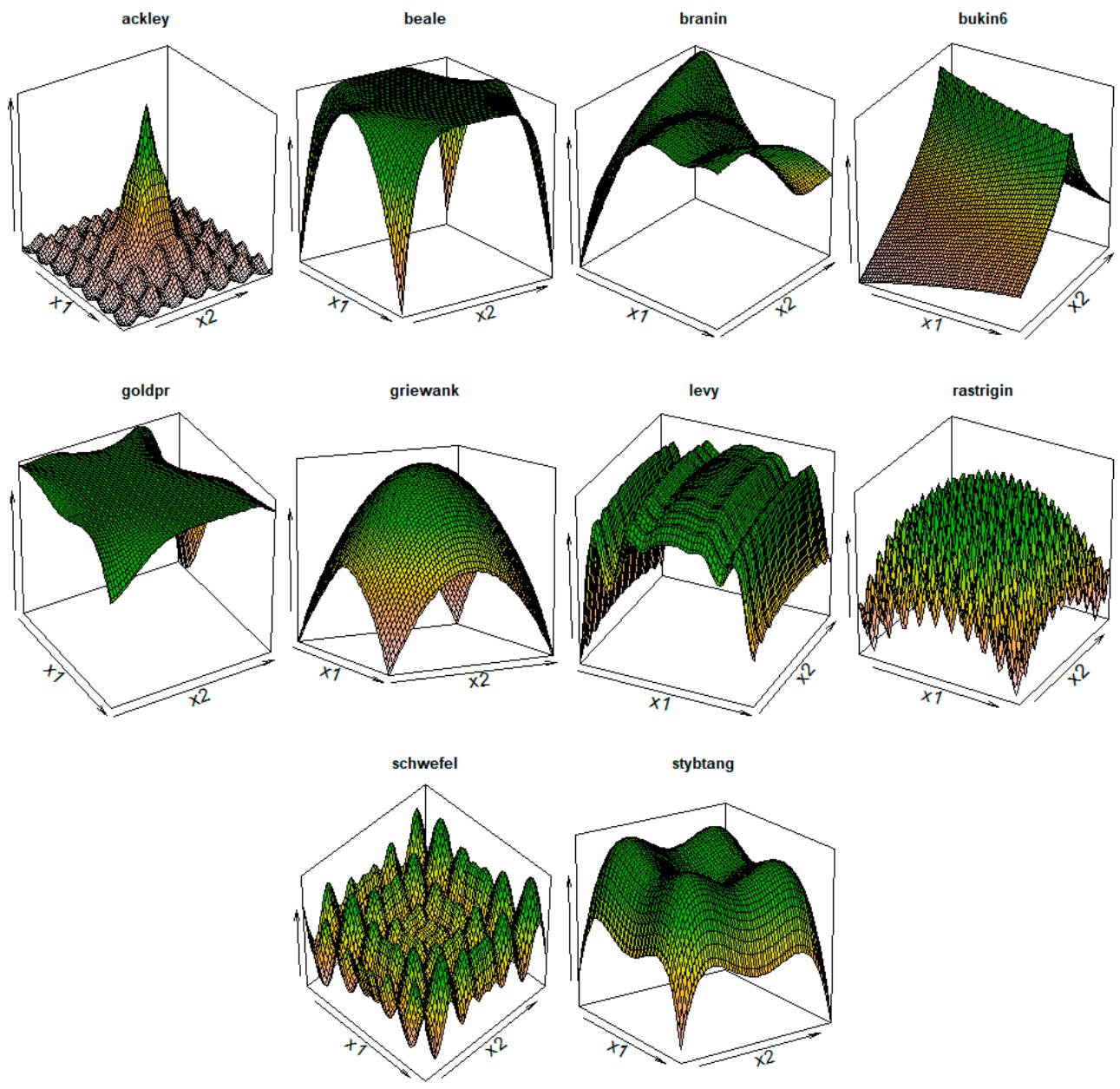

**Figure A1.** The 10 test problems considered in this study.

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
