# Peer review of "Explaining Exploration–Exploitation in Humans"

_2504-2289, doi:10.3390/bdcc6040155_

Round 1
Reviewer 1 Report
I congratulate the author on this fine work, I wish you success.
Below are some minor notes on the manuscript:-
- There is a mixture between the literature and the introduction, Preferably the introduction serves the purpose of leading the reader from a general subject area to a particular field of research. the introduction should introduce your topic and aims and gives an overview of the paper.
- The author should make sure that volume, pagination, and DIO number are mentioned in the reference's bibliography.
Author Response
COMMENT 1. There is a mixture between the literature and the introduction, Preferably the introduction serves the purpose of leading the reader from a general subject area to a particular field of research. the introduction should introduce your topic and aims and gives an overview of the paper.
Reply: thank you for the comment (also raised by another reviewer). We have moved some contents from "motivation" to "related works" with the aim to make both the two sections more coherent and improve readability of the manuscript.
COMMENT 2. The author should make sure that volume, pagination, and DIO number are mentioned in the reference's bibliography.
Reply: we have checked and updated information. We have also added doi (if available). Thank you for the suggestion.
Reviewer 2 Report
I am really grateful to review this manuscript. In my opinion, this manuscript can be published once some revision is done successfully. This study used 39 participants and explainable artificial intelligence (decision tree) to develop a synthetic human player and compare its performance to its Bayesian optimization counterpart in ten tasks. I would argue that this is a rare achievement. However, it can be noted that the attitude of the participant (major predictor of his/her performance) would be affected by his/her age, education and gaming experience but this study did not include these predictors. I would like to suggest the authors to address this issue in Discussion.
Author Response
Thank you so much the appreciation. We are proud that the topic resulted so interesting.
Moreover, thank you for your precious comment. In the revised version of the manuscript we have included, in the Conclusions, the limitation you have suggested.
Reviewer 3 Report
I found this manuscript very interesting in terms of comparing human abilities in decision-making and machine learning performance. The manuscript is well-written and easy to understand. The results show that human performs better in some instances than the machine learning-based approaches which depend upon Average Cumulative Reward (ACR) which is a measure of stress and the test function used. In my opinion, it would be better if the authors could explain why different test functions are performing at different levels of accuracy while dealing with the same input parameters.
Author Response
Thank you so much for the appreciation. We are glad that our workresulted so interesting.
Moreover, thank you so much for your precious comment. In the revised version of the manuscript, we have slightly extended the section 2.2 to refer to the test functions reported in the appendix, remarking that their differences in terms shape clearly affect the final performance of both humans and Bayesian Optimization.
Reviewer 4 Report
Summary
In the article " Explaining exploration-exploitation in humans" The article proposes a comparison between various models for explaining exploration-exploitation in humans. However, there are several concerns that the reviewer would like to point out.
Quires
1. There are numerous grammar errors and typos. Extensive editing of the English language and style is required.
2. In section 1 there is no sentence structure or flow. The author should revise it in order to make it more concise and coherent.
3. There is a need to expand on the concept of limit in equation 7, particularly in the term bound.
4. The article presents the problem in general, followed by a general description of the methods, then by a description of the results obtained with the different methods. The figures 5,6,7 need to be explained in their context.
5. Considering the overall concept, it is quite appealing and admirable. Nonetheless, the article needs to be revised for clarity and coherence to make it easier to read.
Author Response
Thank you for your precious comments. We have carefully considered them in revising our manuscript. As follows, we report our reply to each of them:
COMMENT 1. There are numerous grammar errors and typos. Extensive editing of the English language and style is required.
Reply: we are sorry for the grammar errors and typos. We have proofread the paper and improved the overall readability.
COMMENT 2. In section 1 there is no sentence structure or flow. The author should revise it in order to make it more concise and coherent.
Reply: thank you for the comment. We have revised the section 1, moving part of the contents from "motivation" to "related works" (as also suggested by another reviewer) to make both the sections more coherent.
COMMENT 3. There is a need to expand on the concept of limit in equation 7, particularly in the term bound.
Reply: we are not sure to have understood this comment. Equation 7 is the distance of a point from the approximated Pareto front. We think the term "bound" used by the reviewer refers to what we have called "threshold". In this case, we have provided the following clarification in the manuscript:
"...from a theoretical point of view, a decision is Pareto optimal if it lays exactly on the Pareto front, however, numerical approximation must be considered in practical experiments. From empirical evaluations, we have decided to set this threshold to 10^-4, which proved to be suitable with respect to both computational approximation and invariance to all the functions’ codomains."
We hope to have correctly understood the comment and provided an acceptable clarification.
COMMENT 4. The article presents the problem in general, followed by a general description of the methods, then by a description of the results obtained with the different methods. The figures 5,6,7 need to be explained in their context.
Reply: thank you for the comment. In the revised version of the manuscript, we have introduced and explained each one of the three mentioned figures, individually in their context.
COMMENT 5. Considering the overall concept, it is quite appealing and admirable. Nonetheless, the article needs to be revised for clarity and coherence to make it easier to read.
Reply: thank you so much for the appreciation and, furthermore, for your precious comments and suggestions. We have carefully considered them, allowing us to improve the overall quality of the manuscript. Thank you, again.